# Performance Evaluation of a BZ COVID-19 NALF Assay for Rapid Diagnosis of SARS-CoV-2

**DOI:** 10.3390/diagnostics13061118

**Published:** 2023-03-15

**Authors:** Woong Sik Jang, Hyunseul Jee, Joon Min Lee, Chae Seung Lim, Jeeyong Kim

**Affiliations:** 1Emergency Medicine, College of Medicine, Korea University Guro Hospital, Seoul 08308, Republic of Korea; 2Departments of Laboratory Medicine, College of Medicine, Korea University, Seoul 02841, Republic of Korea; 3Departments of Laboratory Medicine, Korea University Guro Hospital, Seoul 08308, Republic of Korea

**Keywords:** COVID-19, SARS-CoV-2, NALF, nucleic acid, lateral flow, rapid, diagnosis

## Abstract

Coronavirus disease (COVID-19) caused by SARS-CoV-2 infection has been a global pandemic for more than two years, and it is important to quickly and accurately diagnose and isolate patients with SARS-CoV-2 infection. The BZ COVID-19 NALF Assay could sensitively detect SARS-CoV-2 from a nasopharyngeal swab because it adopts both a loop-mediated isothermal amplification and lateral flow immunochromatography technology. In this study, a total of 389 nasopharyngeal swab samples, of which 182 were SARS-CoV-2 PCR positive and 207 were negative samples, were recruited. Compared to the Allplex™ SARS-CoV-2 Assay, the BZ COVID-19 NALF Assay showed 95.05% sensitivity and 99.03% specificity for detecting SARS-CoV-2. The concordance rate between the BZ COVID-19 NALF Assay and Allplex™ SARS-CoV-2 Assay was 97.69%. The turnaround time of the BZ COVID-19 NALF Assay is only about 40~55 min. The BZ COVID-19 NALF Assay is an accurate, easy, and quick molecular diagnostic test compared to the conventional PCR test for detection of SARS-CoV-2. In addition, the BZ COVID-19 NALF Assay is thought to be very useful in small size medical facilities or developing countries where it is difficult to operate a clinical laboratory.

## 1. Introduction

Coronavirus disease (COVID-19) caused by SARS-CoV-2 infection has been a global pandemic for more than two years [1,2]. Globally, there have been 298,915,721 confirmed cases of COVID-19, including 5,469,303 deaths, reported to the WHO by 7 January 2022 [3]. The primary route of human-to-human infection of SARS-CoV-2 occurs through respiratory droplets released by infected individuals. The transmission rate of COVID-19 is 3–4 and the fatality rate is 1.4% [4,5]. Many patients infected with SARS-CoV-2 develop respiratory illness, such as fever, cough, fatigue, and sputum production, and recover without special treatment [4,6]. However, COVID-19 is still a threatening disease to the people around the world, because some patients develop severe symptoms such as difficulty breathing and confusion, and they may even die [7,8]. 

In order to control the outbreak of COVID-19, fast and accurate diagnosis and isolation of patients with suspected SARS-CoV-2 infection is important [9]. Since the WHO published guidelines for diagnosing COVID-19, various diagnostic methods have been developed. Many molecular diagnostic methods, antigen detection methods, and antibody detection methods are used in clinical laboratories. Real-time polymerase chain reaction (RT-qPCR) is the most widely used method for detecting SARS-CoV-2 by amplifying viral-specific genes, such as RNA-dependent RNA polymerase (RdRp), Nucleocapsid (N), Envelope (E), and Spike (S) [10]. SARS-CoV-2 antigen detection is used as a rapid diagnostic test to detect the viral antigens during viral replication. Antibody tests can be used to determine whether people have immunity against SARS-CoV-2 infection [11]. Although rapid diagnostic tests (RDT) are less sensitive than PCR, antigen and antibody tests are easy to operate and portable, making them suitable for point of care testing (POCT) [12]. 

Among these diagnostic methods, the gold standard method for detecting SARS-CoV-2 is RT-qPCR due to its high sensitivity and high specificity [13,14]. However, RT-qPCR has limitations in that it takes about 3 h of test time, and requires an experienced technician and expensive test equipment, although it is the most accurate test method [15]. Several antigen tests have disadvantages of low sensitivity despite their advantages of being cheap and convenient. For example, LIAISON SARS-CoV-2 antigen test (Diasorin, Italy) showed 49.7% sensitivity and 100% specificity [16]. Antibody tests are more suitable for monitoring SARS-CoV-2 infection than for diagnosis because of the fatal limitation of being difficult to diagnose patients in the early stages of infection [17].

Nucleic acid lateral flow (NALF) assay is a fast and convenient in vitro diagnostic test that qualitatively detects specifically labeled nucleic acid. This assay combines the principles of nucleic acid amplification and the lateral flow immunochromatography method [18]. Various PCR methods have been used for nucleic acid amplification. The lateral flow immunochromatography method has been widely used in clinical practice as a rapid immunoassay to detect viral antigens and antibodies. NALF analysis consists of nucleic acid extraction from the sample, nucleic acid amplification, and amplicon detection by lateral flow assay using colorimetric labels [19]. Recently, various NALF assays combining isothermal amplification and lateral flow assay have been reported because the isothermal amplification method is very fast in terms of time compared to RT-qPCR. [20]. These NALF assays can be used to detect SARS-CoV-2 in a POCT because it does not require time-consuming agarose gels or the expensive optical instrument to perform PCR.

In this study, we aimed to demonstrate clinical utility by evaluating the performance of the BZ COVID-19 NALF Assay. The performance of the BZ COVID-19 NALF assay was compared with the Allplex™ SARS-CoV-2 Assay (Seegene, Republic of Korea) which was approved for in vitro diagnostics by Ministry of Food and Drug Safety and CE-IVD. The final purpose of this study was to evaluate whether SARS-CoV-2 could be detected quickly and at a low cost while showing high sensitivity and specificity using the BZ COVID-19 NALF Assay in areas with insufficient medical infrastructure.

## 2. Materials and Methods

### 2.1. Sample Collection and Study Design

In order to evaluate the clinical performance of the BZ COVID-19 NALF Assay, which was developed based on previous reported SARS-CoV-2 RdRP and N gene LAMP primer set [21], 389 nasopharyngeal swab samples were collected from patients suspected with COVID-19 who visited the Korea University Guro Hospital from March 2020 to August 2021. 

The presence or absence of SARS-CoV-2 was determined by synthesizing clinical features, radiological findings, and laboratory test results. SARS-CoV-2 infection can be strongly suspected when the patient had SpO_2_ < 94% on room air at sea level, a ratio of arterial partial pressure of oxygen to fraction of inspired oxygen (PaO_2_/FiO_2_) < 300 mmHg, a respiratory rate > 30 breaths/min, lung infiltrates > 50%, or evidence of lower respiratory disease. If the Allplex™ SARS-CoV-2 Assay was negative despite a suspected SARS-CoV-2 infection, the additional test was performed with the nCoV Real-Time Detection kit (SD Biosensor, Korea). If the nCoV Real-Time Detection Kit result was positive in a patient suspected with SARS-CoV-2 infection, the patient was finally diagnosed as COVID-19 positive, regardless of the Allplex™ SARS-CoV-2 Assay result. SARS-CoV-2 was detected in 182 samples, whereas it was not detected in 207 samples. 

In this study, RNA was extracted from clinical samples using AdvanSure™ E3 system (LG Chem, Seoul, Republic of Korea) and Microlab STARlet IVD (Hamilton Company, NV, USA) according to the manufacturer’s manual. 

To evaluate the performance of a BZ COVID-19 NALF Assay, the limit of detection, sensitivity, specificity, positive predictive value, and negative predictive value of the BZ COVID-19 NALF Assay were compared to that of the Allplex™ SARS-CoV-2 Assay. 

Residual samples and RNAs were stored at −70 °C. This study was approved by the Institutional Review Board of Korea University Guro Hospital, Seoul, Republic of Korea (IRB No. 2021GR0479). 

### 2.2. BZ COVID-19 NALF Assay

The BZ COVID-19 NALF Assay consists of loop-mediated isothermal amplification (LAMP) to amplify the RNAs of SARS-CoV-2 in samples and lateral flow immunochromatography to detect the amplified RNAs. First, RNA was extracted from clinical samples and the N gene and RdRp gene in the sample were amplified through LAMP. Then, if the N gene or RdRp gene is amplified, this was confirmed when a red line appeared through lateral flow assay. 

For LAMP, the Master Mix was prepared with 10 μL of 2 × 1-step RT-LAMP Mix and 5 μL of COVID-19 NALF primer Mix. A total 15 μL of Master Mix was pipetted into the NALF tube. Each 5 μL of RNA extracted from clinical samples was added to NALF tube and mixed 2–3 times. The NALF tube was placed on a heating block (Beijing HiYi Technology, Beijing, China) at 60 °C for 30 min. After LAMP, the lateral flow assay kit was placed it on a clean, flat surface. Six drops (180~200 µL) of buffer were added to the LAMP product. After putting a stopper on the NALF product to which the buffer was added, 3 drops (180 µL) of the NALF product were instilled into the sample well. After 10 min, the results were analyzed with the naked eye without equipment. 

The BZ COVID-19 NALF Assay has one control line and two test lines (T1, T2). When the RdRp gene and N gene of sample were amplified, red bands appeared on the T1 and T2 lines, respectively. Test results were interpreted by combining the presence or absence of these three lines. When a red band appeared on the control line (C) and the test line T1 or T2, it was interpreted as “SARS-CoV-2 Detected”. If a red band appeared on only control line (C), it was interpreted as “SARS-CoV-2 Not detected”. If a red band did not appear on the control line (C), the test result was invalid. In these cases, the test kit was discarded and the test was performed again (Figure 1).

### 2.3. Allplex™ SARS-CoV-2 Assay

To validate the performance of the BZ COVID-19 NALF Assay, RT-qPCR testing was performed using the Allplex™ SARS-CoV-2 Assay, which was approved by CE-IVD and the Ministry of Food and Drug Safety (MFDS) in Korea. The Allplex™ SARS-CoV-2 Assay was able to detect SARS-CoV-2 by targeting the E gene, N gene, RdRp gene, and S gene. The CFX96 Touch Real time PCR detection System (Bio-Rad, CA, USA) was used to identify the amplified viral RNA. 

The PCR mixture was prepared with 14 μL of 2019-nCoV Reaction Solution, 6 μL of RTase Mix, and 0.5 μL of Internal control A. Next, 20 μL of the prepared PCR mixture was dispensed into each well of the real-time PCR reaction tube. A total of 10 μL of RNA extracted from the sample was dispensed into each well of the real-time PCR reaction tube in which the PCR mixture was dispensed. Furthermore, 10 μL of PC and NC were dispensed into the corresponding PC and NC wells. Reaction tubes were centrifuged at low speed for a few seconds and placed in the CFX96 Touch Real time PCR detection System. The thermocycling conditions in the PCR machine were set as follows: reverse transcription at 50 °C for 15 min, initial denaturation at 95 °C for 3 min, 5 cycles of pre-amplification at 95 °C for 5 s and 60 °C for 40 s, and 40 cycles of amplification at 95 °C for 5 s and 60 °C for 40 s. Amplified fluorescence signals were detected through FAM, JOE, and CY5 channels and analyzed. 

The cut-off values of Ct values for the E gene, N gene, RdRp gene, S gene, and internal control were 40. If the E gene, N gene, RdRp gene, S gene, and internal control were all positive, it was interpreted as “SARS-CoV-2 Detected”. If the E gene, N gene, and RdRp/S gene were negative and only the internal control was positive, it was interpreted as “SARS-CoV-2 Not detected”. If the results of the E gene, RdRp/S gene, and N gene did not match, it was interpreted as “Indeterminate”. If the internal control was negative, the test result was invalid and the test was performed again. 

### 2.4. Limit of Detection

In order to evaluate the limit of detection (LOD) of the BZ COVID-19 NALF Assay, the SARS-CoV-2 standard strain was provided from the National Culture Collection for Pathogens (NCCP). SARS-CoV-2 no. 43346, wild-type strain was distributed and spiked in normal nasopharyngeal samples. These spiked samples serially diluted 10-fold from 10^3^ PFU/mL to 10^−3^ PFU/mL to evaluate detection limit. These diluted samples were repeatedly tested three times for each concentration.

### 2.5. Statistical Analysis

Test results using the BZ COVID-19 NALF Assay were evaluated in comparison with RT-qPCR results using the Allplex™ SARS-CoV-2 Assay which was a clinically approved method for detecting SARS-CoV-2. The diagnostic performance of the BZ COVID-19 NALF Assay was evaluated for clinical sensitivity, specificity, positive predictive value (PPV), negative predictive value (NPV), and concordance rate. The concordance rate between the BZ COVID-19 NALF Assay and RT-qPCR using Allplex™ SARS-CoV-2 Assay was calculated using inter-rater agreement statistics (kappa). A *p* value of <0.05 was considered statistically significant. All statistical analyses were performed using SPSS for Windows (version 22.0; IBM Corporation, NY, USA).

## 3. Results

### 3.1. Limit of Detection of the BZ COVID-19 NALF Assay

For SARS-CoV-2 spiked samples folded from 10^3^ PFU/mL to 10^−3^ PFU/mL, the detection limit of the T1 test line in the BZ COVID-19 NALF Assay is the same as that of the Allplex™ SARS-CoV-2 assay, which is 10^−1^ PFU/mL (Figure 2). However, the detection limit of the T2 test line in the BZ COVID-19 NALF Assay is two steps higher than that of the Allplex™ SARS-CoV-2 assay, which is 10^1^ PFU/mL.

### 3.2. Clinical Performance

A comparative analysis of SARS-CoV-2 detection by the BZ COVID-19 NALF Assay and the Allplex™ SARS-CoV-2 Assay was performed. Of 182 SARS-CoV-2 positive samples tested by the BZ COVID-19 NALF Assay and Allplex™ SARS-CoV-2 Assay, 173 and 180 samples were detected, respectively. Of 207 SARS-CoV-2 negative samples tested by the BZ COVID-19 NALF Assay and Allplex™ SARS-CoV-2 Assay, 205 and 207 samples were not detected, respectively (Table 1). 

The sensitivities of the BZ COVID-19 NALF Assay and Allplex™ SARS-CoV-2 Assay were 95.05% and 98.90%, respectively. The specificities of the BZ COVID-19 NALF Assay and Allplex™ SARS-CoV-2 Assay were 99.03% and 100.00%, respectively. The positive predictive values of the BZ COVID-19 NALF Assay and Allplex™ SARS-CoV-2 Assay were 98.86% and 100.00%, respectively. The negative predictive values of the BZ COVID-19 NALF Assay and Allplex™ SARS-CoV-2 Assay were 96.73% and 99.04%, respectively. The concordance rate between the BZ COVID-19 NALF Assay and Allplex™ SARS-CoV-2 Assay was 97.69%. 

Table 2 summarizes the BZ COVID-19 NALF Assay results according to the threshold cycle (Ct) of the Allplex™ SARS-CoV-2 Assay. In Table 2, the Ct value is indicated as the Ct value for the RdRp/S gene among the three Ct values of the RdRp/S, N, and E genes of the Allplex™ SARS-CoV-2 Assay. In Ct values of the RdRp/S genes of the Allplex™ SARS-CoV-2 Assay for positive clinical samples, mean, median, and standard deviation were 24.73, 23.21, and 6.27, respectively. In this study, the Ct values of the E gene and N gene in the Allplex™ SARS-CoV-2 Assay were not described.

However, discrepant results between the BZ COVID-19 NALF Assay and the Allplex™ SARS-CoV-2 Assay occurred in 9 of total 389 clinical samples (Table 3). In seven positive clinical samples, the Allplex™ SARS-CoV-2 Assay detected positively, but the BZ COVID-19 NALF Assay did not detect positively. For two negative clinical samples, the Allplex™ SARS-CoV-2 Assay correctly diagnosed them as negative, but the BZ COVID-19 NALF Assay gave false positive results.

The turnaround time (TAT) of the BZ COVID-19 NALF Assay was about 40~55 min. It took about 5–20 min for nucleic acid extraction, about 30 min for LAMP, and about 5 min for lateral flow assay.

## 4. Discussion

COVID-19 is still a global pandemic, and the damage is staggering. Unlike some developed countries where vaccination rates are high and herd immunity has been achieved, control of COVID-19 is still necessary in many countries. Accurate and quick diagnosis and treatment are important for the proper management of COVID-19, and a medical system that can support it is needed. Accordingly, many diagnostic methods for COVID-19 have been developed, and new diagnostic methods are being developed according to the unmet needs of medical staff.

Although there are several methods of diagnosing COVID-19, RT-qPCR is currently considered the standard method. Because RT-qPCR is currently considered the gold standard test for the diagnosis of COVID-19 due to its high sensitivity and specificity, various commercial RT-qPCR assays have been developed [22]. However, RT-qPCR is difficult to use in small laboratories or developing countries because it requires a thermocycler that can change various temperatures from cycle to cycle, skilled technicians, and other expensive instruments [23]. Various test methods are being developed because there is a need for a rapid and accurate diagnostic method that can detect the presence of SARS-CoV-2 virus in clinical samples. A one-step single-tube nested quantitative real-time PCR method was developed for the rapid detection of the SARS-CoV-2 pathogen in clinical samples [24]. A field-effect transistor (FET)-based biosensor was developed for quick detection of SARS-CoV-2 infection in patient samples [25]. A few reverse transcription LAMP tests for detecting SARS-CoV-2 virus were developed for POCT [26,27]. 

In one study, optimized LAMP assay showed a good analytical sensitivity comparable to RT-qPCR with less than 10 copies per reaction, while another LAMP assay showed an unsuitable limit of detection for clinical use [27,28]. In general, LAMP assays had the advantage that they did not require expensive equipment such as a thermocycler. However, conventional LAMP assays showed inferior diagnostic performance compared to RT-qPCR [29,30]. A serological test using lateral flow immunochromatography was easy and used as a rapid test method for detecting SARS-CoV-2, but it has the disadvantages of low sensitivity and long incubation period [31,32,33]. In this study, we tested the clinical performance of the BZ COVID-19 NALF Assay which combines the LAMP and lateral flow assay techniques. In the detection limit test, the T1 test line (for RdRP gene) of the BZ COVID-19 NALF Assay showed the same detection limit (10^−1^ PFU/mL) of the Allplex™ SARS-CoV-2 Assay. In addition, the BZ COVID-19 NALF Assay showed sensitivity of 95.05% and specificity of 99.03% for detecting SARS-CoV-2. The concordance rate between the BZ COVID-19 NALF Assay and Allplex™ SARS-CoV-2 Assay was 97.69%. These results showed that the ability of the BZ COVID-19 NALF Assay to detect SARS-CoV-2 was comparable to the Allplex™ SARS-CoV-2 Assay, which is a commercial RT-qPCR assay. The sensitivity of the BZ COVID-19 NALF Assay was superior to those previously reported for SARS-CoV-2 antigen tests (64–76%) or antibody tests (79–93%) [34,35]. Therefore, the BZ COVID-19 NALF assay is considered suitable for use in vitro diagnostic detection of SARS-CoV-2 in clinical settings.

Out of 182 positive clinical samples, 7 discrepant results occurred between the BZ COVID-19 NALF Assay and the Allplex™ SARS-CoV-2 Assay. Seven positive samples were detected in the Allplex™ SARS-CoV-2 Assay but not in the BZ COVID-19 NALF Assay. The mean, median, and standard deviation Ct values by Allplex™ SARS-CoV-2 Assay in these seven samples were 36.28, 36.65, and 2.81, respectively. Since these Ct values were close to the cut-off value of the Allplex™ SARS-CoV-2 Assay, it was considered a weak positive sample with a small amount of SARS-CoV-2. Therefore, the BZ COVID-19 NALF Assay should consider the possibility of false negative results when the viral load of SARS-CoV-2 in the sample is small. 

Of the 207 negative clinical samples, 2 discrepant results occurred between the BZ COVID-19 NALF Assay and the Allplex™ SARS-CoV-2 Assay. Two negative samples were detected in the BZ COVID-19 NALF Assay but not in the Allplex™ SARS-CoV-2 Assay. Two false positive results occurred in the BZ COVID-19 NALF Assay, but the cause was not identified. Since no specific cause has been identified, the possibility of a random error was considered.

The BZ COVID-19 NALF Assay is a new test that combines the advantages of LAMP and lateral flow assay. With the LAMP method, the BZ COVID-19 NALF Assay can amplify nucleic acids of SARS-CoV-2 without requiring a thermocycler, and results can be obtained within 1 h using a heating block. This assay also uses a lateral flow immunochromatography method, which allows it to be easily read with the naked eye, similar to a pregnancy self-diagnosis kit. This makes it easier to diagnose COVID-19 in a clinical laboratory, without requiring experienced technicians or expensive test equipment. Overall, the BZ COVID-19 NALF Assay represents a promising new approach to COVID-19 testing that could make it more accessible and affordable for a wider range of people.

## 5. Conclusions

The BZ COVID-19 NALF Assay exhibits a high sensitivity of 95.05% and specificity of 99.03%, which is comparable to that of the Allplex™ SARS-CoV-2 Assay. Moreover, with a short turnaround time of approximately 40–55 min, the BZ COVID-19 NALF Assay offers rapid and accurate detection of SARS-CoV-2, which is particularly useful in limited-resource settings or small medical facilities where access to expensive diagnostic equipment or skilled technicians is limited.

## Figures and Tables

**Figure 1 diagnostics-13-01118-f001:**
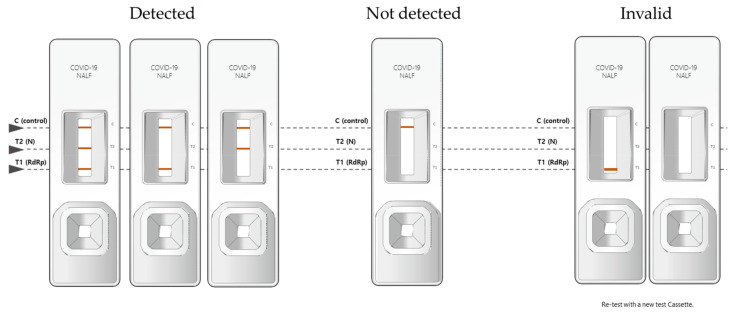
Interpretation criteria of the BZ COVID-19 NALF Assay.

**Figure 2 diagnostics-13-01118-f002:**
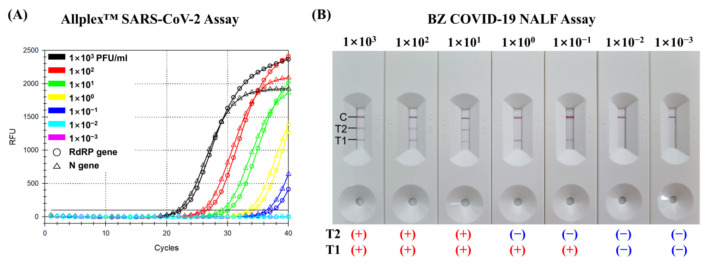
Comparison of the LOD of the Allplex^TM^ SARS-CoV-2 Assay and BZ COVID-19 NALF Assay. The limit of detection (LOD) for two assays was tested with RNA samples of SARS-CoV-2 wild-type (NCCP no. 43346) ranging from 10^3^ to 10^−3^ PFU/mL. (**A**) LOD of Allplex^TM^ SARS-CoV-2 Assay. (**B**) LOD of BZ COVID-19 NALF Assay. (+) and (−) indicate positive and negative results of the reaction, respectively.

**Table 1 diagnostics-13-01118-t001:** Results of the Allplex™ SARS-CoV-2 Assay and BZ COVID-19 NALF Assay.

SARS-Cov-2	Allplex™ SARS-CoV-2 Assay		BZ COVID-19 NALF Assay	
	Detected	Not Detected	Detected	Not Detected
Positive samples	180	2	173	9
Negative samples	0	207	2	205

**Table 2 diagnostics-13-01118-t002:** Test results according to the threshold cycle (Ct) of the Allplex™ SARS-CoV-2 Assay.

Clinical Samples	Threshold Cycle (Ct)	Allplex™ SARS-CoV-2 Assay	BZ COVID-19 NALF Assay
Positive Samples (*n* = 182)	10.00–19.99	53 Detected	53 Detected
	20.00–29.99	85 Detected	84 Detected/1 Not detected
	30.00–39.99	42 Detected	36 Detected/6 Not detected
	No amplification	2 Not detected	2 Not Detected
Negative Samples (*n* = 207)	No amplification	207 Not detected	2 Detected/205 Not detected

**Table 3 diagnostics-13-01118-t003:** Discrepant results and Ct values between the BZ COVID-19 NALF Assay and Allplex™ SARS-CoV-2 Assay.

Sample	Allplex™ SARS-CoV-2 Assay	BZ COVID-19 NALF Assay
1	Detected (Ct = 29.81)	Not detected
2	Detected (Ct = 36.57)	Not detected
3	Detected (Ct = 37.79)	Not detected
4	Detected (Ct = 37.69)	Not detected
5	Detected (Ct = 36.19)	Not detected
6	Detected (Ct = 36.65)	Not detected
7	Detected (Ct = 39.27)	Not detected
8	Not detected	Detected
9	Not detected	Detected

## Data Availability

The authors declare that all related data are available from the corresponding author upon reasonable request.

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
