# Peer review of "Performance Evaluation of a BZ COVID-19 NALF Assay for Rapid Diagnosis of SARS-CoV-2"

_diagnostics, 2023, doi:10.3390/diagnostics13061118_

Round 1

Reviewer 1 Report

My main comments are:

Line 52-53. Please add the cost of this test: "LIAISON SARS-CoV-2 antigen test (Diasorin, Italy) showed 49.7% sensitivity and 100% specificity" ... with a cost of ...

Line 69-70. Is this a previous result? "The BZ COVID-19 NALF Assay was a fast and convenient in vitro diagnostic test that qualitatively detects SARS-CoV-2 from nasopharyngeal swab"

Line 81. "In order to evaluate the clinical performance of the BZ COVID-19 NALF Assay", no information is provided of the kit. Is it commercially available? Is it yours? Please include proper references.

Line 101. Is RNA extraction simple and cheap? Does it require an expensive equipment or expensive reagents. A technician must perform this step? Nothing is mentioned in this respect. 

Line 114-15. Use µL instead of uL, and separate units from dimensions

Line 175-178. The statement of "no line appeared" seems odd to me. Perhaps the redness was reduced. Is it possible to determine a true detection limit? Run a densitometric analysis if possible. 

Line 199-200. The values of 24.73, 23.21 and 6.27 correspond to which assay?

Line 242-243. "LAMP assays showed lower limit of detection for SARS-CoV-2 than RT-qPCR". Provide more data that support this. What is the SARS-CoV-2 limit of detection using RT-qPCR. 

Line 252-253. "was superior to previously reported performance of SARS-CoV-2 antigen tests or antibody tests". Please provide data to support this statement. 

Line 268-269. "In two samples, neither the BZ COVID-19 NALF Assay nor the Allplex™ SARS-CoV-2 Assay failed to detect SARS-CoV-2", they both succeed then?

There are also several grammar and writing errors (have a native English speaker or a professional service reviewing the manuscript):

These NALF assays can be used to detect amplicons in a POCT because it does not require

all samples were tested by BZ COVID-19 NALF Assay and Allplex™ SARS-CoV-2 Assay, respectively.

BZ COVID-19 NALF Assay were compared to Allplex™ SARS-CoV-2 Assay.

but BZ COVID-19 NALF Assay was detected.

SARS-CoV-2 virus

which was existing licensed RT-qPCR assays.

These performance of the BZ COVID-19 NALF

Assay were considered sufficient 

The Allplex™ SARS-CoV-2 Assay was detected for SARS-CoV-2, but 

but BZ COVID-19 NALF Assay was detected.

These SARS-CoV-2 detection performance

Author Response

We appreciate your interest and detailed comments on our manuscript.

We have revised the manuscript based on the comments of the reviewers and marked all changes we made.

During the revision process, we added one author (Hyunseul Jee) who performed additional LOD experiments.

The missing funding from Korea University (grant number: K2225641) was also added.

Sincerely, 

Jeeyong Kim

Reviewer 2 Report

Interesting and relevant topic is discussed. But, some revisions to manuscript are needed. Overall, the article sounds like an advertisement. One of the author CS Lim declare a financial relationship with Biozentech, therefore, all doubts about hidden advertising need to be cleared. It is not correct to write product prices. Firstly this is a kind of advertisement, secondly these are prices for your country only.

Section Materials and Methods. Figure 1 is not needed and it is not appropriate.

Section Results. Lines 169-173; 213-215 -The texts are for Materials and Methods

Section Discussion. This section needs to be re-edited. It is not arranged well. The prices should be removed.

Lines 256-272 The text need clarification. You wrote that 182 were the positive samples, 180 of which  were positive by Allplex Assay and 173 by NALF. There are two positive samples left. How are these two samples proven to be positive?

Author Response

(The authors gave the same response as above.)
